# Prevalence of Lower Urinary Tract Symptoms in Children with Attention-Deficit/Hyperactivity Disorder: Comparison of Hospital and Population-Based Cohorts of 13,000 Patients

**DOI:** 10.3390/jcm11216393

**Published:** 2022-10-28

**Authors:** Teng-Kai Yang, Wei-Yi Huang, Ya-Jun Guo, Yu-Fen Chen, Hong-Chiang Chang, Kuo-How Huang

**Affiliations:** 1Division of Urology, Department of Surgery, Yonghe Cardinal Tien Hospital, New Taipei 23148, Taiwan; 2School of Medicine, College of Medicine, Fu-Jen Catholic University, New Taipei 242062, Taiwan; 3Institute of Health and Welfare Policy, National Yang-Ming University, Taipei 112, Taiwan; 4Department of Healthcare and Medical Care, Veterans Affairs Council, Taipei 23742, Taiwan; 5Department of Psychiatry, Taipei City Hospital, Taipei 110309, Taiwan; 6Department of Nursing, Kang-Ning Junior College of Medical Care and Management, Taipei 11485, Taiwan; 7Department of Urology, National Taiwan University Hospital, Taipei 10025, Taiwan

**Keywords:** attention deficit hyperactivity disorder, lower urinary tract symptoms

## Abstract

**Background:** This study investigates the prevalence of lower urinary tract symptoms (LUTS) in school-age children with Attention-Deficit/Hyperactivity Disorder (ADHD) based on hospital-based and population-based cohorts. **Methods:** The hospital-based sample comprised 42 children with ADHD and 65 without ADHD aged 6–12 years. Voiding dysfunction was assessed by the Dysfunctional Voiding Scoring System (DVSS) questionnaire. We compared the baseline data, DVSS score, and uroflowmetry between the two groups. For the population-based cohort in the national insurance database, we included 6526 children aged 6–12 years, whose claims record included the diagnosis of ADHD, and another 6526 control subjects matched by gender and age. We compared the presence of LUTS diagnosis codes between the two groups. **Results**: Our results showed that, for the hospital-based cohort, the mean total DVSS score and the proportion of significant LUTS in children in the ADHD group were significantly higher than in subjects in the non-ADHD group. The DVSS subscales showed that the item “I cannot wait when I have to pee” item was significantly higher in the ADHD group (1.62 ± 1.17 vs. 0.90 ± 1.09, *p* = 0.002). For the population-based cohort, children with ADHD had a significantly higher likelihood of storage symptoms (5.53% vs. 2.91%, *p* < 0.001) and enuresis (3.28% vs. 1.95%, *p* < 0.001) compared with those of the no ADHD group. **Conclusions:** Children with ADHD have a higher prevalence of significant LUTS, especially storage symptoms and enuresis, than children without ADHD. The observed correlations between ADHD and LUTS provided the supporting evidence to evaluate the concomitant voiding dysfunction in children with ADHD.

## 1. Introduction

Attention-Deficit/Hyperactivity Disorder (ADHD) is an occurring pediatric and adolescent neurodevelopmental disorder that affects around 5–10% of school-age children [1]. The genetic deficiency and specific brain dysfunction may play a role in the pathogenesis of ADHD. ADHD is associated with a variety of other diseases [2]. The prevalence of Lower urinary tract symptoms (LUTS) in children has increased in recent years. Approximately one-third of children worldwide have symptoms associated with voiding problems [3,4]. Due to modernized lifestyle, the prevalence of LUTS in children increases because of inappropriate voiding habits and toilet training [4,5]. Epidemiologically, the ratio of LUTS in school-aged children is around 2 to 20 percent, mostly urgency, intermittency, urinary incontinence and enuresis, which affect social limiting and decrease health related quality of life of the children and the parents [4,6].

Emerging evidence indicates that voiding dysfunction or nocturnal enuresis are associated with ADHD in children [7,8,9,10,11,12,13]. Problems with bedwetting and other bladder control symptoms adversely affect the sense of well-being and psychosocial function in children [14,15,16]. Children with ADHD have been reported to be more susceptible to lower urinary tract symptoms (LUTS) such as nocturia or enuresis [8,9,10,13,15]. Nevertheless, studies on the prevalence of LUTS or voiding dysfunction in children with ADHD are limited.

Few studies compared hospital-based and population-based cohorts directly. The strength and power of the correlation between potential confounders for disease or bothersome symptoms differed according to controls selection. A weaker or negative estimation with hospital-based controls may exist compared with population-based controls because of selection bias [17,18].

This study examines the prevalence of LUTS in school-age children with ADHD, investigates whether children with ADHD are more likely than their peers to have LUTS, and compares the results of the hospital- and population-based data.

## 2. Materials and Methods

### 2.1. Hospital-Based Sample

#### 2.1.1. Participants

Between January 2013 and December 2015, we enrolled 42 children aged between 6 and 12 years with confirmed diagnoses of ADHD from the child and adolescent psychiatric clinic in a tertiary referral center in Taiwan. Another 65 children without ADHD were enrolled from pediatric clinic in the same hospital for other non-LUTS related complaints. ADHD was diagnosed according to DSM-IV criteria as the existence of disease symptoms for more than 6 months continuously and in two different environments (at home and school) without any organic causes. The validated Chinese version of the SNAP-IV was written by the parent(s) of each child [19,20]. Children with (1) other genetic problems, such as seizure and mental retardation, and organic central nerve diseases, such as seizures, concurrent neurologic or urogenital disorders; (2) previous pelvic surgery; or (3) the present use of drugs known to interfere with bladder or sphincter function, such as anticholinergics, diuretics, antidepressants and antipsychotics, were excluded from the study. The data of demographics, medical history, the existence of a sleep problems, perinatal history and developmental history were acquired from the parents or medical records.

#### 2.1.2. Dysfunctional Voiding Symptom Scale (DVSS) Questionnaire

LUTS in children were defined based on the terminology of the International Children’s Continence Society [3]. The assessment of LUTS was based on uroflowmetry and the validated Chinese version of the DVSS questionnaire validated for quantifying voiding symptoms in children [21]. The validated Chinese version of the DVSS questionnaire was provided to the parents of each child to evaluate LUTS. The validated DVSS consists of 10 questions that evaluate enuresis, voiding frequency, urgency, daytime incontinence and constipation. The score was from 0 to 3 (0 = almost never, 1 = less than half the time, 2 = about half the time, 3 = almost every time) for each question with a maximum total score of 30 (most severe symptoms) [22,23]. Dysfunctional voiding was defined as “inappropriate constriction of the urethral sphincter or pelvic floor during voiding in neurologically normal individuals”. Therefore, it is rationally that children with dysfunctional voiding would have voiding and storage symptoms during voiding, just as shown in the International Prostate Symptom Score [24]. In the Chinese validation study, the 10 questions were classified into three categories, including four items (Q1, Q2, Q6, and Q7) related to storage symptoms. Another category included difficult elimination syndrome because there is a connected association between bladder and bowel impairment. Constipation in children may cause decreased bladder contractility or bladder outlet obstruction and then lead to voiding difficulty. It is biologically plausible to categorize Q3, Q4, and Q8 as voiding symptoms. DVSS 7 was a cutoff for significant LUTS [21]. Each child was instructed to drink water until a strong desire to void was achieved, then they were instructed to complete the uroflowmetry.

### 2.2. Study Sample from Nationwide Health Insurance Database

#### 2.2.1. National Health Insurance Research Database (NHIRD)

The National Health Insurance Bureau of Taiwan (NHI) has collected records of all inpatient and outpatient medical benefit claims for nearly the entire population of Taiwan since the inception of its single-payer NHI program in 1995. The entire data collection is known as the NHIRD [25].

#### 2.2.2. Longitudinal Health Insurance Database 2005 (LHID2005)

This study used a dataset, the Longitudinal Health Insurance Database (LHID) 2005, collected from the NHIRD. The LHID2005 collected all the original claims data of 1,000,000 recipients enrolled in 2005, randomly sampled from the 2005 registry for recipients of the NHIRD. There are approximately 25.6 million individuals in this registry. There was no significant difference in the gender distribution between the patients in the LHID2005 and the original NHIRD [26], which represent the whole population of Taiwan [27,28].

#### 2.2.3. LUTS

LUTS diagnosis codes were defined by the International Classification of Diseases, Ninth Revision, Clinical Modification (ICD-9-CM) as follows: Voiding symptoms contained codes of voiding difficulty (788.1), urinary retention (788.2, 788.20), incomplete emptying (788.21), postvoid dribbling (788.35), weak urine stream (788.6), splitting of the urinary stream (788.61), and slowing of the urinary stream (788.62). Storage symptoms contained frequency (788.4, 788.41), polyuria (788.4, 788.42), incontinence (788.3, 788.30–788.39, 625.6), nocturia (788.43) bladder hypertonicity (596.51), nocturnal enuresis (788.36), nocturia (788.43), and urinary urgency (788.63). Subjects whose diagnoses had concurrent codes of urinary tract infection (590.1, 590.2, 590.8, 590.9, 595.0, 595.9, 599.0, 996.64) were excluded from the sample.

#### 2.2.4. Concomitant Anomalies

Concomitant anomalies or congenital disabilities were identified by ICD-9-CM codes 740–759 in all study subjects and were also included in the analysis. Then, we analyzed the presence of any of specific diagnoses of LUTS in study sample during the study period, 2013 to 2015. 

#### 2.2.5. Study Sample

In the NHIRD-based sample, we included 6526 children aged 6–12 years, whose claims record included the diagnosis of ADHD (ICD code: 314.00, 314.01) during the period. Another 6526 control subjects were randomly selected whose claims records did not contain any diagnosis of ADHD. The control subjects were matched to the ADHD patients by gender, age, and having claims records in the year of the case subject’s initial ADHD diagnosis.

This institutional-based study was approved by the Research Ethics Committee of National Taiwan University Hospital. The national insurance database used in this study was deidentified; therefore, the institutional review board waived informed consent from the enrolled patients.

### 2.3. Statistical Analysis

The numerical data were described by means and standard deviations or medians and ranges. The categorical data were expressed as counts and percentages. The numerical and the categorical data were compared with the Student’s *t*-test and chi-square test. Two-sided tests were used, and a *p*-value of less than 0.05 was statistically significant. All data in the present study were analyzed with commercial statistical software (SPSS version 20.0 for Windows, SPSS Inc., Chicago, IL, USA).

## 3. Results

### 3.1. The Hospital-Based Sample

The baseline data of the study sample are presented in Table 1. A total of 107 school-age children, 42 with ADHD and 65 without ADHD, were enrolled. The mean age of children in the ADHD and the no ADHD groups were 8.00 ± 2.20 years and 8.26 ± 2.00 years, respectively (*p* = 0.55). There were 14 (33.3%) girls and 28 (66.6%) boys in the ADHD group; 18 girls (27.7%) and 47 (72.3%) boys in the no ADHD group (*p* = 0.54). The maternal age of the subjects with ADHD was significantly younger than those without ADHD (37.9 vs. 40.1 years, *p* = 0.01). There were no significant differences between the ADHD and the no ADHD groups regarding parental age, prematurity, household income, maternal education, maternal marital status, sibling number, type of delivery, and perinatal history. The DVSS scores with subscale, flow rate, and voided volume of study subjects are shown in Table 2. The total DVSS of children with ADHD was significantly higher than that of the no ADHD group (8.24 ± 6.09 vs. 6.09 ± 4.65, *p* = 0.042). The subscale, “I cannot wait when I have to pee,” was significantly higher in the ADHD group (1.62 ± 1.17 vs. 0.90 ± 1.09, *p* = 0.002). In addition, the voided volume of the uroflowmetry study was significantly lower in the ADHD group (73.8 ± 3.58 vs. 100 ± 7.57, *p* = 0.045). The sleep quality score of the ADHD group was worse than the no ADHD group (2.88 ± 2.00 vs. 1.91 ± 1.86, *p* = 0.013). Although constipation (14.2% vs. 7.7%) and enuresis (26.1% vs. 18.5%) were more frequently seen in children with ADHD than those without ADHD, the differences were not statistically significant. For the hospital-based cohort in Table 3, there were significantly more LUTS in children with ADHD (47.6% vs. 21.5%, *p* = 0.045) than in children without ADHD. The storage, voiding symptoms, and enuresis were comparable for both groups. 

### 3.2. The Nationwide, Population-Based Sample

A total of 6526 children with the diagnosis of ADHD and another 6526 age- and gender-matched control subjects without ADHD diagnoses were enrolled. Demographically, Table 4 showed that the families of school-aged children with ADHD had significantly higher insurable monthly wages and lived in areas of a higher urbanization level than those without ADHD (*p* < 0.001). In the claim records, regarding associated anomalies, children with ADHD had a higher risk of concomitant anomalies. Among them, when compared with those of the no ADHD group, children with ADHD had more musculoskeletal (2.7% vs. 1.5%, *p* < 0.001) and neurological (1.3% vs. 0.6%, *p* < 0.001) anomalies. According to Table 3, children with ADHD had a significantly higher likelihood of storage symptoms in their claim’s records than those of the no ADHD group (5.5% vs. 2.9%, *p* < 0.001). Additionally, children with ADHD were more likely to have LUTS diagnoses (5.8% vs. 3.3%, *p* < 0.001) and nocturnal enuresis (3.28% vs. 1.94%, *p* < 0.001).

## 4. Discussion

Previous studies have indicated that voiding dysfunction and enuresis are more prevalent in children with ADHD [8,9,10,13,15]. Consistently, our study showed that children with ADHD have significantly more prevalent LUTS and significantly more severe voiding dysfunction than children without ADHD for hospital-based and population-based cohorts [10,12]. For significant LUTS in the population-based cohort, children with ADHD showed a significantly higher proportion of predominantly storage symptoms (*p* < 0.01), which showed only borderline significance in the hospital-based cohort (*p* = 0.06). 

For reasons of representativeness, utilizing a population-based cohort is admitted as being more acceptable than that of a hospital-based cohort. One conceivable reason for the differences showed between the two cohorts was Berkson’s bias. The hospital-based cohort may have been easier to be admitted for exposure of interest like enuresis, and it was the severity of the disease that led to hospital visits. This would bias the results for a correlation toward the null hypothesis with the hospital-based cohort and away from the null hypothesis utilizing the population-based cohort [17].

The exact theories to explain the associations between ADHD and voiding dysfunction remain unclear. Decreased brain stem inhibition led to failure to detect the bladder signals and lower responsiveness to a full bladder were thought to be involved in the linkage [29,30]. In addition, inappropriate central adrenergic stimulation in children with ADHD causes sphincter overactivity with subsequent dysfunctional voiding may be involved in the linkage between ADHD and voiding dysfunction [10]. The behavioral pattern, such as hyperactivity and inattention in children with ADHD, may play an important role in the appropriate response to the desire to urinate. 

The correlation between ADHD and enuresis was mentioned in previous studies [8,11,12,13,14,15,16,31]. Voiding dysfunction, or LUTS, represents up to 40% of pediatric urology consults [32]. Studies on the national- or population-based prevalence of voiding dysfunction enuresis in school-aged children are scarce [33]. A previous study in a sample of 1136 school-aged children enrolled in National Health and Nutrition Examination Surveys from 2001 to 2004 showed that the prevalence of enuresis was 4.45% in the whole sample, 4.02% in children with ADHD, and 2.88% in children without ADHD [33]. The prevalence in boys was significantly greater than in girls. ADHD was associated with enuresis (odds ratio 2.88). However, only 36% of the enuretic children had received health services for enuresis. The author concluded that the evaluation of LUTS should routinely include in the assessment of ADHD and vice versa.

Our study sample comprised children aged between 6 and 12 years. The diagnosis of LUTS in the nationwide population-based sample mainly depended on the claims data of the health insurance rather than an accurate medical history reported by the parents or physical examinations and urinalysis. However, diagnoses of voiding dysfunction in the hospital-based sample were based on the validated DVSS questionnaires [22,34] and uroflowmetry and tended to be more reliable. Our results from the two study samples showed compatible results. In the nationwide, population-based sample, children with ADHD had more prevalent LUTS (5.79% vs. 3.36%) and enuresis (3.28% vs. 1.95%) than those without ADHD. The prevalence of healthcare-seeking enuresis was 3.28% in the ADHD group and 1.94% in the control group. The lower prevalence rate of enuresis in our study subjects may be attributed to a proportion of subjects with enuresis who did not seek medical treatment.

In the hospital-based database, children with ADHD had a higher DVSS score and higher subscale, “I cannot wait when I have to pee,” and less voided volume of the uroflowmetry study. In addition, the sleep fragmentation of the ADHD group was worse than that of the no ADHD group. The institution-based sample showed that children with ADHD had a higher total DVSS score, higher probability of the subscale (cannot wait to pee), and lower voiding volume than those in the no ADHD group. The healthcare-seeking prevalence of LUTS in children with ADHD was 5.8% compared with 3.3% in those without ADHD. Meanwhile, children with ADHD had a significantly higher risk of storage symptoms and nocturnal enuresis.

The strengths of the present study are as follows: First, previous studies were mostly conducted in western countries and had a cross-sectional design. Our study provides the 6-year observation of healthcare-seeking prevalence of LUTS in an Asian cohort. Second, the NHI covers over 99% of Taiwan’s 23 million population, and the NHI databases are representative of the general population. Moreover, NHIRD datasets are not designed for research purposes and can minimize selection bias. In addition, the insurance claims data inevitably included some diagnostic errors. All claimed records of the NHIRD are subject to a quality-control process that includes cross-comparison with medical chart information. In this way, some diagnostic errors in the raw claims data are rectified during the quality-control process.

Our study has some limitations. First, we investigated the presence and severity of voiding at outpatient clinic by single measurements for DVSS and uroflowmetry at one time-point, which may limit the reliability of the research in the hospital-based sample. Due to invasiveness for the children, we did not collect sufficient data such as pressure flow study, bladder sonography for a greater bladder wall thickness, and post-void residual volume, so further urodynamic approaches, and outcomes with long follow up durations are warranted in the future. Second, some study subjects with the diagnosis of ADHD in the nationwide health insurance database were diagnosed by primary care physicians rather than by a comprehensive assessment by psychiatrists. Moreover, the prevalence of LUTS in the population-based sample was healthcare-seeking prevalence, which may be lower than the data of previous studies conducted by interviews. It is worth noting that a significant proportion of children with enuresis or voiding dysfunction did not seek medical treatment. Nevertheless, the healthcare-seeking LUTS is clinically significant to some degree. Lastly, ADHD is composed of three types of symptoms: attention deficit, hyperactivity, and impulsivity. It is necessary to conduct prospective studies on the prevalence of voiding dysfunction in school-age children with different ADHD symptoms. 

## 5. Conclusions

School-age children with ADHD have a higher prevalence of LUTS, especially bothersome storage symptoms and voiding dysfunction than those without ADHD. Child healthcare providers should routinely screen for LUTS or voiding dysfunction in school-age children with ADHD.

## Figures and Tables

**Table 1 jcm-11-06393-t001:** The comparison of baseline characteristics of school-age children with Attention-Deficit/Hyperactivity Disorder (ADHD) versus no ADHD in the hospital-based cohort.

	No ADHD (n = 65)	ADHD (n = 42)	*p* Value
Age (yrs)	8.26 ± 2.00	8.00 ± 2.20	0.55
Paternal Age (yrs)	41.8 ± 4.28	41.1 ± 6.27	0.54
Maternal Age (yrs)	40.1 ± 3.74	37.9 ± 4.57	0.011
Birth weight (gm)	2991 ± 544	3128 ± 494	0.23
No. of siblings (person)	0.97 ± 0.61	1.05 ± 0.73	0.55
Gender F/M (%)	18/47 (27.7/72.3)	14/28 (33.3/66.6)	0.53
Premature (<37 wks) (%)	6(9.4)	7 (17.5)	0.22
Maternal education (%)			0.41
College	26 (40.6)	12 (28.6)	
High school	36 (56.3)	29 (69.0)	
Elementary school	2 (3.1)	1 (2.4)	
Household Incomes (%)			0.07
Low	6 (9.5)	7 (17.9)	
Middle	25 (39.7)	21 (53.8)	
High	32 (50.8)	11(28.3)	
Marital status (%)			0.10
Intact marriage	58 (92.1)	36 (85.7)	
Separated	0	3 (7.1)	
Divorced	5 (7.9)	3 (7.1)	
Pt. with Siblings	53 (81.5)	32 (76.2)	0.50
Type of Deliver (%)			0.91
Normal Spontaneous Delivery	42 (66.7)	27 (65.9)	
Emergent Cesarean Section	9 (14.3)	7 (17.1)	
Elective Cesarean Section	12 (19.0)	7 (17.1)	
Perinatal insult (%)	3 (5.3)	1 (2.9)	0.58
Newborn jaundice (%)	20 (87.0)	18 (62.1)	0.05

Household incomes: Low means monthly incomes <30,000 NT dollars; moderate means incomes between 30,000 and 80,000 NT dollars; high means incomes ≥80,000 NT dollars. Numeric data are expressed as Mean ± SD and compared with *t*-test. Categorical data are expressed as number (percentage) and compared with chi-square test.

**Table 2 jcm-11-06393-t002:** The comparison of Dysfunctional Voiding Symptom Scale (DVSS) and flow rates of school-age children, with ADHD versus no ADHD in the hospital-based cohort.

	No ADHD (n = 65)	ADHD (n = 42)	*p* Value
	Mean ± SD	Mean ± SD	
Total DVSS	6.09 ± 4.65	8.24 ± 6.09	0.042
Subscale			
Wet underwear	0.49 ± 0.90	0.74 ± 0.94	0.18
Soak underwear	0.67 ± 0.98	0.95 ± 1.15	0.18
No daily bowel movement	0.80 ± 0.89	0.86 ± 0.84	0.73
Push to have bowel movement	0.66 ± 0.89	0.90 ± 0.96	0.18
Pee 1–2 times/day	0.41 ± 0.73	0.69 ± 1.07	0.11
Hold pee	1.25 ± 1.02	1.31 ± 1.22	0.78
Cannot wait	0.90 ± 1.09	1.62 ± 1.17	0.002
Push to pee	0.17 ± 0.49	0.29 ± 0.74	0.34
Hurt when pee	0.16 ± 0.41	0.33 ± 0.85	0.16
Stressful events	0.97 ± 1.41	1.24 ± 1.50	0.39
Peak flow rate, mL/s	15.2 ± 7.16	14.8 ± 5.24	0.78
Voided volume, mL	100 ± 7.57	73.8 ± 3.58	0.045
Total sleep quality score	1.91 ± 1.86	2.88 ± 2.00	0.013
Enuresis (%)	12 (18.5)	11 (26.1)	0.29
Constipation (%)	5 (7.7)	6 (14.2)	0.14

Numeric data are expressed as mean and SD and compared with *t*-test. Categorical data are expressed as number (percentage) and compared with chi-square test.

**Table 3 jcm-11-06393-t003:** The comparison of significant LUTS in school-age children with ADHD versus no ADHD, in population-based and hospital-based cohorts.

	Population-Based Cohorts			Hospital-Based Cohorts		
	No ADHD (%)	ADHD (%)	*p* Value	No ADHD (%)	ADHD (%)	*p* Value
	n = 6526 (100)	n = 6526 (100)		n = 65 (100)	n = 42 (100)	
Significant Lower urinary tract symptoms	219 (3.36)	378 (5.79)	<0.01	14 (21.5)	20 (47.6)	0.045
Storage symptoms predominate	190 (2.91)	361 (5.53)	<0.01	12 (18.5)	17 (40.5)	0.06
Voiding symptoms predominate	29 (0.44)	17 (0.26)	0.08	2 (3.08)	3 (7.14)	0.36
Enuresis	127 (1.95)	214 (3.28)	<0.01	12 (18.5)	11 (26.1)	0.29

Numeric data are expressed as mean and SD and compared with *t*-test. Categorical data are expressed as number (percentage) and compared with chi-square test.

**Table 4 jcm-11-06393-t004:** The baseline characteristics in school-age children with ADHD versus no ADHD, in the nationwide, population-based cohort.

	No ADHD (%)	ADHD (%)	*p* Value
	n = 6526 (100)	n = 6526 (100)	
Age (years)	8.13 ± 1.92	8.13 ± 1.93	1.00
Gender			1.00
Girls	1343 (20.6)	1343 (20.6)	
Boys	5183 (79.4)	5183 (79.4)	
Insurable monthly wage (New Taiwan Dollor)			<0.01
<24,800	3529 (54.1)	2734 (41.9)	
≥24,800	2997 (45.9)	3792 (58.1)	
Urbanization level of residence			<0.01
Metropolitan	2695 (41.3)	3514 (53.9)	
Urban	2688 (41.2)	2251 (34.5)	
Rural	1143 (17.5)	761 (11.7)	
Anomalies			
genitourinary tract	31 (0.48)	45 (0.69)	0.11
cryptorchidism	47 (0.72)	64 (0.98)	0.11
hypospadias	7 (0.11)	8 (0.12)	0.80
Preterm	92 (1.41)	114 (1.75)	0.12
Chromosomal	2 (0.03)	12 (0.18)	0.01
Musculoskeletal	129 (1.98)	229 (3.51)	0.01
gastrointestinal	31 (0.48)	52 (0.80)	0.02
cardiovascular	296 (4.54)	386 (5.91)	0.20

Numeric data are expressed as mean and SD and compared with *t*-test. Categorical data are expressed as number (percentage) and compared with chi-square test.

## Data Availability

The datasets used and/or analysed during the current study are available from the corresponding author on reasonable request.

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
