# Peer review of "Prevalence of Lower Urinary Tract Symptoms in Children with Attention-Deficit/Hyperactivity Disorder: Comparison of Hospital and Population-Based Cohorts of 13,000 Patients"

_jcm, 2022, doi:10.3390/jcm11216393_

Round 1
Reviewer 1 Report
Dear Authors, thank you for giving me the opportunity to review your manuscript. There are some comments that I would like to make with the intention of improving it:
1. Please, provide sufficient background by defining and specifying the LUTS group of symptoms in the Introduction.
2. Line 170: Please, clarify the study period 1997-2012. It is confusing as your hospital-based cohort was enrolled in 2013-2015, and your LHID registry is of 2005.
3. I recommend to change the title:
Prevalence of Lower Urinary Tract Symptoms in children with Attention-Deficit/Hyperactivity Disorder: comparison of hospital and population-based cohorts of 13 000 patients
4. Please provide the version of the DVSS in the Supplementary Materials, or refer to the one that has been used (line 112)
5. Please, specify it have been Taiwan psychiatric and pediatric clinics (lines 94 and 95)
6. Please, change throughout the text to Attention-Deficit/Hyperactivity Disorder (ADHD) (title, lines 36, 65 etc.)
7. Please, delete excessive full stops (lines 55, 222, 277, etc.)
8. Please, add a space before the reference and put a full stop after it throughout the text (lines 67, 69, 71, 73, 82, 100, 126, 134, 153, 238, 249 etc.)
9. Please, reconcile tenses of the verbs (lines 83-84)
10. Missing the first paragraph (lines 100-106)
11. Missing the second inverter commas (lines 120-122)
12. Missing comma (line 162)
13. Please, unify your “P” value throughout the text, as there are three types of it (P, p and p)
14. Missing space (lines 240, 241)
15. Minor Comments:
Line 95: complaints
Line 97: delete “; or”
Line 135: delete “then”
Line 288: study
Reviewer 2 Report
Dear Author
I have reviewed the manuscript entitled "Prevalence of Lower Urinary Tract Symptoms in Children with Attention Deficit-Hyperactivity Disorder, The comparison of hospital and population-based cohorts". I have some comments below;
1. Abstract section; line 55; please remove the point after blanket.
2. Introduction section is too short, please add more literature information
3 Why the study enrolled between 2013 and 2015? Please explain. I think you must enroll the participants at least 5 years ago.
4. I think only a questionnaire is not enough to determine uninhibited contraction of the bladder. You must perform the urodynamic evaluation of the patient to diagnose uninhibited contractions. you must redesign your study
5. table 4; correct the sentences (lower instead of lowe)
best regards
Round 2
Reviewer 1 Report
Thanks for addressing all the comments
Author Response
Thank you for your valuable comments
Reviewer 2 Report
Dear Author
The references that you added to introducton is not related with your introduction section. It is related with your method and you must add it to the material and method section. Your introduction section is still not enough and must be support ith the latest references. Your data is too old. you must add new participants or you can give long term outcomes of this data.
Sincerely your
